# miR-10 and Its Negative Correlation with Serum IL-35 Concentration and Positive Correlation with STAT5a Expression in Patients with Rheumatoid Arthritis

**DOI:** 10.3390/ijms23147925

**Published:** 2022-07-18

**Authors:** Agnieszka Paradowska-Gorycka, Anna Wajda, Ewa Rzeszotarska, Tomasz Kmiolek, Barbara Stypinska, Ewa Dudek, Katarzyna Romanowska-Prochnicka, Piotr Syrowka

**Affiliations:** 1Department of Molecular Biology, National Institute of Geriatrics, Rheumatology and Rehabilitation, 02-637 Warsaw, Poland; anna.wajda@spartanska.pl (A.W.); ewa.rzeszotarska@spartanska.pl (E.R.); tomasz.kmiolek@spartanska.pl (T.K.); barbara.stypinska@spartanska.pl (B.S.); ewa.dudek@spartanska.pl (E.D.); 2Department of Connective Tissue Diseases, National Institute of Geriatrics, Rheumatology and Rehabilitation, 02-637 Warsaw, Poland; katarzyna.prochnicka@gmail.com; 3Department of Pathophysiology, Warsaw Medical University, 02-091 Warsaw, Poland; 4Rheumaorthopedics Clinic and Polyclinic, National Institute of Geriatrics, Rheumatology and Rehabilitation, 02-637 Warsaw, Poland; piotr.syrowka@spartanska.pl

**Keywords:** miRNA, rheumatoid arthritis, transcriptional factors

## Abstract

Circulating free-cell miRNAs are increasingly important as potential non-invasive biomarkers due to the easy accessibility of clinical materials. Moreover, their epigenetic role may provide insight into the mechanisms of pathogenesis. Nevertheless, these aspects are mostly studied in the area of oncological diseases. Therefore, this research aimed to find the potential association of selected miRNAs in serum with the expression of Th17/Treg transcription factors and clinical features in RA patients. Accordingly, experiments was conducted on rheumatoid arthritis (RA), osteoarthritis (OA) and healthy subjects (HC). Analysis of miRNAs level in serum was performed using LNA miRNA PCR assays. mir-10 was detected only in RA patients. Furthermore, its expression was correlated with IL-35 serum concentration and the mRNA level of *STAT5a* in whole blood in RA. Additionally, a tendency of the raised level of miR-10 was noted in RA patients with high activity disease. miR-326 was significantly upregulated in RA patients with rheumatoid factor presence. In HC the correlation between miR-26 and IL-21 serum levels and expression of *SMAD3* have been found. In OA patients, correlations between miR-126 and *HIF1* expression and between miR-146 and *RORc* have been noted. The differential association of transcription factor expression with serum miRNA levels may be important in the diagnosis and progression of RA and OA.

## 1. Introduction

Rheumatoid arthritis (RA) and osteoarthritis (OA) are both multifactorial diseases with unknown etiology, affecting cartilages and joints. RA and OA are connected with enlarged morbidity and mortality [1,2,3]. These diseases share parallel pathophysiological pathways like cell senescence, agglomeration (in joints and the skeletal tissues) of activated immune cells, enlarged bone remodeling. RA is described by chronic synovitis, osteoclast activation, and subchondral bone resorption, while in OA destruction of the cartilage extracellular matrix (ECM) is escorted by osteophyte formation, subchondral bone remodeling, and, with the progression of the OA, synovial inflammation [4]. However, the pathogenesis of RA and OA remains elusive [5].

CD4+ T cells have an essential role in the immune regulation. Th17 cells play an important role in inflammation, autoimmune response, and tumor immunity. Cytokines, responsible for the induction and maintenance of Th17 cells, stimulate signal transducer and activator of transcription 3 (STAT3), which is crucial in the differentiation of Th17 cells, which in turn are important in the immunopathology of RA and other autoimmune diseases [4,6,7]. STAT3 signaling pathway impairs responses induced by TGF-β via the direct interplay between SMAD family member 3 (SMAD3) and STAT3 [8]. SMAD3 is a key factor that regulates the production of inflammatory cytokines as well as the activation of T cells [9]. SMAD3 has a key role in the downregulation of T cells, enlarging the expression of forkhead box protein 3 (FOXP3), which is a key step in Treg cells’ differentiation [10]. FOXP3, crucial in Treg development, is turned on by STAT5. Additionally, STAT5 counteract the activation of STAT3 [11]. Suppressor of cytokine signaling 1 (SOCS1) is a negative controller of cytokines and modulates cell activation [12]. SOCS1 is essential for the immunosuppressive function of Treg cells because the absence of SOCS1 causes the production of inflammatory cytokines, such as IFN-γ and IL-17 by these T cells [13]. SOCS1 may participate in the Th17 cells’ development by inhibiting the IFN-γ antagonistic effect on the differentiation of Th17 cells [13]. Furthermore, the suppressor of cytokine signaling 1 impairs activation of STAT3, mediated by IL-10 [14]. SOCS1 protein is involved in the pathogenesis of diseases with the autoimmune compound [13].

MicroRNAs (miRNAs, miRs) are small, non-coding RNAs that regulate gene expression at post-transcriptional level. MiRNAs are crucial controllers of cellular processes like apoptosis, differentiation and maturation. Furthermore, miRs have a pivotal role in homeostasis, modulation of development and the role of immune cells like Treg and Th17 cells. MiR-155 and miR-146a overexpression was detected in regulatory T cells from RA patients [15,16,17]. Loss of the function of Treg cells, caused by the dysregulation of miRs, is demonstrated by the breakdown of immune tolerance as well as impairment of immune suppressive functions [15].

Present study is a continuation of our earlier research where we determine whether there is a correlation between transcription factors expression and Th17/Treg ratio [18,19]. Based on our study, we observed that SMAD3 and STAT3 could be potential diagnostic biomarkers for RA as well as that in RA patients with DAS-28 > 5.1, SOCS1, and STAT3 mRNA levels were lower in Treg cells and higher in Th17 cells. Our earlier studies [19] have also shown that the expression of miR-26 was positively correlated with *SMAD3*, *STAT3*, and *SOCS1* expression, and the expression of miR-155 was positively correlated with *STAT3* expression in RA Th17 cells. In RA Treg cells positive correlations were observed between miR-26 and *SOCS1*, miR-31 and *SMAD3*, and miR-155 and *SMAD3/SMAD4*. This study aimed to recognize the potential impact of the miRNAs’ expression profiles, which are the expression levels of miR-10, -23a, -24, -26a, -27a, -31, -100, -126, -146a, -155, -210, -326, and -451a in serum on the RA onset and phenotype, and also on the selected transcriptional factors, SMAD3, STAT3, and SOCS1 in RA, OA, and healthy controls (HCs). In addition, we studied the relationship between SMAD3, STAT3, SOCS1 and cytokine serum levels as well as examined cell-free circulating miRNAs, which could distinguish RA, OA, and HCs. Expression analyses of miRNAs were performed in the same group of patients as in our previous study, but the difference here is that we obtained miRNA from serum, whereas before it was obtained from earlier sorted Th17 and Treg cells. We chose miRNAs, responsible for regulating the gene expression of transcriptional factors impacting the balance of Th17/Treg cells, which are essential in RA pathogenesis [19].

## 2. Results

### 2.1. miRNA Expression in Serum

miRNAs, also examined in the present study, have been reported to show altered expression in T cells, such as Treg and Th17 cells [19]. Considering free-cell miRNA, also known as circulating miRNAs, as potential biomarkers, their analysis in serum would be of interest due to the easy accessibility of clinical materials. miR-31 and miR-100 expression levels in serum were not detected in our study population. In healthy subjects (HCs), miR-326 and miR-27 were below quantification, and miR-26, miR-146a, and miR-210 were at very low levels. We also observed that miR-10 was detected only in the serum of RA patients.

In the next step, we investigated a correlation analysis to indicate a potential direction for further functional research on the mechanisms of miRNA regulation. Significant, negative correlation has been noted between miR-10 and miR-155 (r = −0.75, *p* = 0.007), miR-10 and miR-23 (r = −0.67, *p* = 0.02), and between miR-126 and miR-326 (r = −0.66, *p* = 0.011) in serum in RA patients (Figure 1).

In OA patients, significant correlation has been observed between miR-26 and miR-126 expression in serum (r = 0.87, *p* = 0.004) and between miR-126 and miR-451 (r = 0.94, *p* = 0.02). Data have also shown a significant, very strong correlation between miR-126 and miR-24 (r = 0.95, *p* = 0.001) and miR-126 and miR-451 (r = 0.97, *p* = 0.0002) in healthy subjects (Figure 2).

### 2.2. miRNA and RA/OA Phenotype

Considering the importance of the miRNA in the pathogenesis of autoimmune diseases as well as their important role as biomarkers, we also analyzed whether examined miRNA expression may have an impact on the RA and OA phenotype. We observed that RA patients with positive rheumatoid factors (RF) were characterized by higher levels of miR-326 in serum in comparison to RF negative RA patients (*p* = 0.046) (Figure 3A). The miR-10 serum levels were higher in RA patients with DAS 28 > 5.1 when compared to RA patients with DAS-28 ≤ 5.1. However, the difference was not significant (*p* = 0.35) (Figure 3B).

The expression of all examined miRNAs and anti-CCP presence in RA was also investigated. We demonstrated that RA patients with anti-CCP presence were characterized by higher levels of miR-26 (*p* = 0.26) and miR-126 (*p* = 0.64) in serum than in RA patients without anti-CCP. However, the difference was not statistically significant (Figure 3C,D).

### 2.3. Cytokine Serum Levels

Cytokines control and regulate a wide range of inflammatory processes, and an imbalance between pro- and anti-inflammatory cytokines leads to the induction of autoimmunity, chronic inflammation, and in consequence joint damage. The concentrations of different, Th17 or Treg-related cytokines (IL-17, IL-21, IL-22, IL-6, IL-10, IL-35, IL-2, IL-23, TGF-β, IFN-γ) in serum was also estimated in our earlier study [19].

In RA patients, we revealed an average correlation between the age of patients and IL-21 serum levels (r = −0.48, *p* = 0.02). Additionally, a significant and high correlation between IL-6 serum levels and the mean value of ESR has been also observed (r = 0.69, *p* = 0.0004) (Figure 4).

In the present study, we also observed that RA patients with an elevated level of the mean value of ESR (42.25 ± 21.91 mm/h) were characterized by about two times higher IFN-γ serum level when to compare with RA patient with normal mean value of ESR (11.40 ± 2.65 mm/h), but the difference was not significant (*p* = 0.25). Moreover, RA patients with high activity of diseases (DAS28 > 5.1) and the RA patients in remission or with mild or moderate disease activity (DAS28 ≤ 5.1) did not differ when compared the analyzed cytokines serum levels. In case of anti-CCP presence, negative RA patients had a higher serum concentration of IL-2 than patients with anti-CCP presence (*p* = 0.01) (Appendix A).

RA patients with RF presence were characterized by significantly lower serum levels of IL-17F (*p* = 0.02), IL-17 (*p* = 0.04), IL-10 (*p* = 0.04), IL-22 (*p* = 0.02), and IL-21 (*p* < 0.001) than RA patients without RF (Figure 5). Serum levels of IL-6 in RA patients with RF and without RF were at similar levels (median concentration 3.6 and 3.3 pg/mL, respectively, data not shown).

### 2.4. miRNA Expression Levels in Serum and Correlation between Transcriptional Factors

The present study revealed a significant correlation between miR-10 serum level and *STAT5a* mRNA level in whole blood in RA patients (r = 0.63, *p* = 0.04) (Appendix A). *STAT5a* expression in whole blood was negatively correlated with miR-26 level in serum in OA patients (r = −0.94, *p* = 0.005) (Appendix A).

In RA patients, a significant negative correlation has been shown between miR-26 serum level and expression of *SMAD3* (r = −0.69, *p* = 0.02)/*FOXP3* (r = −0.84, *p* = 0.004)/*SMAD4* (r = −0.64, *p* = 0.04)/*HIF1* (r = −0.65, *p* = 0.03) (Appendix A). Whereas in HC, miR-26 level in serum was positively correlated with *SMAD3* expression in whole blood (r = 0.94, *p* = 0.04) (Appendix A).

In OA patients, miR-126 expression in serum has been correlated with *HIF1a* mRNA level in whole blood (r = −0.88, *p* = 0.03) (Appendix A), and miR-146a serum level with *RORc* expression in whole blood (r = −0.71, *p* = 0.14) (Appendix A).

### 2.5. miRNA Expression Levels in Serum and Correlation between Cytokine Levels

A negative significant correlation has been shown between miR-10 expression in serum and IL-35 level (r = −0.85, *p* = 0.006; Figure 6) and between miR-326 expression in serum and IL-35 level in RA patients (r = −0.85, *p* = 0.006; Figure 6). In HCs, a significant correlation has been noted between miR-26 and IL-21 levels (r = 0.70, *p* = 0.03). In OA patients, no correlation between cytokine concentration in serum and serum level of analyzed miRNA has been revealed.

## 3. Discussion

Over the last few years, we have witnessed a growing interest in miRNAs and their control of the immune system’s function. MiRNAs play a crucial role in pathogenic defense, differentiation and signaling in the immune system. Deregulation of miRNAs related to the immune system causes long-term inflammation that is representative of chronic inflammatory diseases [20]. Nevertheless, depending on the methods used and the biological material, the studies on extracellular RNA, including miRNA in biological fluids may lead to discrepancies in research and complicate the identification of reliable and relevant miRNA markers. MiRNA profiling in biofluidic samples is challenging, mainly due to the lack of method standardization from the isolation stage through the further analysis steps, e.g., broad range of reference genes, sometimes incorrect normalization to exogenous spike-in [21,22].

In rheumatoid arthritis, inflammation of synovial tissue causes irreversible joint damage [20]. MiR-146a, miR-155, miR-23, miR-451, miR-126, and others have been documented in RA tissues [20,23,24,25].

In the present study, we focused on miRNAs in serum using modified miRNA by locked nucleic acid (LNA™) technology, which ensures higher affinity and specificity than standard oligonucleotide probes [26]. Additionally, we checked the level of selected cytokines in serum which are involved in the possible immunological disbalance in patients with autoimmune diseases. The present study was designed to (1) determine differences in the expression profile of selected miRNAs in serum of patients with RA, OA, and HC; (2) find potential correlations between miRNAs; (3) find relationships/associations between miRNA expression and serum cytokine concentration/expression levels of selected transcription factors in whole blood; (4) note associations with clinical parameters.

We have not detected miR-31 and miR-100 in the serum of our entire study population, and in healthy controls, miR-27 and miR-326 were below quantification, while miR-26 and miR-146 and miR-210 were at very low levels. What is more, we revealed that miR-10 was detected only in serum of RA patients. Researchers found that miR-10a expression was meaningfully downregulated in RA (in comparison to OA patients), and this caused upregulation of IL-6 and other inflammatory cytokines. However, these studies were conducted on human synovial sarcoma cell line (SW982) and in the fibroblast-like synoviocytes (FLSs) [27,28]. What is more, we exposed negative correlations between miR-10 and miR-155, and between miR-10 and miR-23 in serum of RA patients. Our previous study did not reveal expression of miR-10a in Treg and Th17 cells (data not shown), therefore present outcomes suggest that lysis of monocytes/macrophages or endothelial cells may be a potential source of miR-10a in the serum of RA patients as one of the key regulators of inflammatory signaling pathway [29,30]. Another source may be an effect of the treatment [31]. Interestingly, the present study shows a negative correlation between miR-10a expression and serum IL-35 concentration and a positive correlation with *STAT5a* mRNA level in whole blood in RA patients. Additionally, we have noted a tendency of raised miR-10a expression in RA patients with high disease activity. However, the difference was not statistically significant and this observation seems to contradict the dogma of the anti-inflammatory effect of miR-10 [32,33]. Nevertheless, the association of miR-10 with STAT5a has been proved by Sun et al. [31]. Authors noted that miR-10a silencing inhibits the phosphorylation of STAT5a. Thus, miR-10a is able to regulate TGF-β/Smad2/STAT3/STAT5a signaling pathway. The lack of miR-10a and very low *STAT5a* expression in Treg cells and/or Th17/Treg disbalance in RA patients [18] may lead to the impaired secretion of IL-35.

IL-35 is a relatively new discovered cytokine, produced mostly by Treg cells via JAK/STAT pathway and its anti-inflammatory or pro-inflammatory effect remains controversial [34,35]. Elevated serum IL-35 level was observed in patients with steroid-sensitive asthma which may suggest the involvement of this cytokine in the corticosteroid enhancing effect in monocytes [36]. A study on Chinese RA patients revealed an association between elevated serum IL-35 levels with disease activity [37]. However, our previous research, conducted on the Caucasian population, did not show a significant difference in serum IL-35 levels between RA, OA and healthy subjects [18]. Furthermore, another study revealed lower serum IL-35 concentration in RA patients than in SLE and APS patients [38]. In the present study, we also observed a correlation between serum IL-35 concentration and miR-326 level in RA patients. Furthermore, RA patients with the presence of rheumatoid factor (RF) had higher levels of miR-326 than RA patients with RF-negative. However, Sui et al. found that, in Chinese RA patients, miR-326 was negatively linked with RF, but this analysis was done in peripheral blood, not in serum [39]. Nevertheless, there is some consistency between our results and Sui’s study. The authors noted a decrease in miR-326 levels after treatment [39]. Thus, it should be noted that in our study, patients were under the treatment. In addition, it is determined that RF-negative patients have a better prognosis in disease course and response to treatment [40]. In OA patients, we did not reveal correlations between miRNAs and cytokines, but we observed a correlation between miR-126 with *HIF1a* and miR-146a with *RORc*. Interestingly, we found a correlation between mir-126 and miR-451 in OA and HC groups but not in RA. In the serum of RA patients, we noted negative correlations between miR-126 and miR-326, whereas in healthy subjects, we observed correlations between miR-126 and miR-24. These different associations may be due to pathological processes and different treatments. However, with small sample sizes, caution must be applied, as the findings might not be transferable to mechanistic relationships of the expression regulation.

We also observed a negative correlation between miR-26 and *SMAD3*, *FOXP3*, *SMAD4* and *HIF1a* in RA patients. Whereas in healthy subjects, the miR-26 serum level was positively correlated only with *SMAD3* expression. Liang et al. proved the antifibrotic effects of miR-26 and demonstrated that miR-26a blocks nuclear translocation of *p*-Smad3 by targeting directly Smad4, which determines nuclear translocation of *p*-Smad2/Smad3 [41]. In the OA group, we noticed a negative correlation between *STAT5a* and miR-26 levels. Our previous study also revealed correlation between miR-26 and *STAT5a* expression in Treg cells in RA but neither in OA nor healthy subjects. However, Treg cells in healthy subjects were characterized by a higher expression of miR-26 than Treg cells in OA. Interestingly we show that in healthy subjects, the expression of this miRNA in Treg cells is higher than in Th17 [19]. In OA patients, we also noticed correlations between miR-26 and miR-126. Despite the antifibrotic properties of miR-26, different studies demonstrated its significance in skeletal muscle differentiation and regeneration [42] or reduction of inflammatory responses [43]. Therefore, it is possible to hypothesize that these roles are associated with a positive correlation between miR-26 and serum IL-21 concentration in healthy subjects. What is surprising is that the IL-21 level was much higher in HCs than in RA and OA. Rasmussen et al. found that in RA at the early stage IL-21 was elevated, however, this analysis was performed in plasma, not in serum [44]. Nevertheless, Kwok et al. and Chen et al. revealed that active RA patients had enlarged serum levels of IL-21, but when patients were in remission after anti-TNF-α therapy, their serum levels of IL-21 were meaningfully decreased [45,46]. Therefore, we hypothesize that the decreased level of IL-21 in RA patients can be explained by the stage of the disease and consequently, the treatment applied to RA patients. Other concentrations of cytokines, e.g., IL-22, IL-6, IL-10, IL-35, and TGF-β, were on similar levels in all groups, including HCs.

Furthermore, we noticed connections between cytokines and clinical parameters. In addition to the above results, we observed in RA a negative correlation between age and IL-21 levels. We also saw the correlation between mean ESR value and IL-6 levels in RA patients. Patients with different disease activities did not differ in cytokines levels. We showed that anti-CCP negative RA patients had higher IL-2 levels than RA anti-CCP positive patients. RA patients with rheumatoid factor had significantly lower levels of IL-17F, IL-17, IL-10, IL-22, IL-21 than RA patients with RF-negative. Additionally, IL-6 levels in RA patients with and without RF were on similar levels. Moreover, RA patients with higher mean ESR values had two times greater IFN-γ values than RA patients with normal ESR values, but this outcome was not significant as well. RA and OA patients had higher IL-2 and IFN-γ levels than HCs.

Unfortunately, the present study was limited in several ways. Due to technical and financial constraints, we could not analyze miRNAs expression in the larger group of patients or more homogeneous groups of patients. Nevertheless, we wanted to present all available clinical data to show the broadest picture of the patients, even if there is inconsistency in the data presented (e.g., lack of Vit D concentration in RA patients). Moreover, nutrition and lifestyle aspects also affect miRNAs and biochemical homeostasis [47,48]. However, these are beyond the scope of the present research. Additionally, the present study was conducted on RA patients with a different scheme of treatment and with different disease activity. Nevertheless, the results shed light on interesting interactions and differences observed in RA patients, but not OA or healthy subjects or occurring in OA and not RA. Therefore, further research is required to determine exactly how miR-10 affects RA development and what is the role of IL-35 on miR-10 expression. Moreover, the association between IL-21 and miR-26 is an important issue for future research.

## 4. Materials and Methods

### 4.1. Study Population

This study was conducted on a group of 23 RA patients, 26 OA patients, and 29 healthy controls (HCs). Analysis of miRNA in serum was conducted on the part of the patients. Patients with RA were recruited from the National Institute of Geriatrics, Rheumatology and Rehabilitation in Warsaw, Poland, and from the Poznan University of Medical Sciences, Poland. The inclusion criteria for RA patients were: all (male and female) RA patients who fulfill the 1987 American College of Rheumatology (ACR) or the 2010 EULAR/ACR criteria for RA, aged ≥ 18 years, and of Polish ethnicity. Patients with OA were recruited from the National Institute of Geriatrics, Rheumatology and Rehabilitation in Warsaw, Poland. OA patients were diagnosed based on characteristic X-ray findings and the absence of features suggestive of inflammatory arthritis and must meet the ACR criteria for OA of the knee. Exclusion criteria for RA and OA patients were: a history of or current other inflammatory rheumatological or autoimmune disorders; malignancy; significant unstable or uncontrolled acute or chronic disease, which could confound the results of the study and/or current active infection.

Demographic and clinical parameters of RA and OA patients are presented in Table 1 and Table 2. In Table 1, we present the clinical data of all study populations, whereas in Table 2 we present clinical data patients and controls used for miRNAs analysis. The patients and controls for miRNA analysis were randomly selected. All data information were collected at the time of the clinical materials sampling.

Healthy subjects were selected from healthy blood bank donor volunteers without a history of autoimmune/inflammatory diseases and cancers. Healthy donors were selected to match the RA patients in age, gender, and ethnicity. RA patients and HCs were from the same geographical area.

Present study was approved by the Ethics Committee of the National Institute of Geriatrics, Rheumatology, and Rehabilitation, Warsaw, Poland (approval protocol number 29 June 2016) and it meets all criteria contained in the 1964 Declaration of Helsinki. All RA patients and healthy subjects gave their written informed consent before enrolment.

### 4.2. miRNA Isolation

Total RNA including miRNA was isolated from 200 ul of serum using miRNeasy Serum/Plasma kit (Qiagen Hilden, Germany; Cat No./ID 217184) according to the manufacturer’s instruction. To standardize samples in the whole experiment the same volume of serum has been taken. UniSp2, UniSp4, UniSp5 miRCURY LNA miRNA PCR Assays were used to control the miRNA isolation process.

### 4.3. miRNA Expression

Analysis of miRNAs level in serum was performed using LNA miRNA PCR assays: hsa-miR-10a-5p, hsa-miR-23a-3p, miR-24-3p, hsa-miR-26a-5p, miR-27a-3p, hsa-miR-31-5p, hsa-miR-100-5p, hsa-miR-126-3p, hsa-miR-146a-5p, hsa-miR-155-5p, hsa-miR-210-5p, hsa-miR-326, and hsa-miR-451a (Qiagen Cat No./QG-339306). Reverse transcription of miRNA was performed with miRCURY LNA RT kit (Qiagen Cat No./ID 339340) according to the manufacturer’s instruction. Quantitative Real-Time PCR was performed with miRCURY SYBR Green PCR Kit (Qiagen Cat No./ID 339347). qRT-PCR was performed on QuantStudio 5 real-time PCR System (Applied Biosystems, Foster City, CA, USA). Each sample was analyzed in duplicate. Using the SNORD48 snRNA PCR primer set, UniRT was taken as housekeeping gene. The relative expression of analyzed miRNAs was calculated by the ΔCt method.

### 4.4. Cytokine Serum Levels

For quantitative determination of cytokine serum levels, the samples from RA and OA patients and healthy subjects were separated from peripheral venous blood at room temperature and stored at −86 °C until analysis. Levels of Il-17A (pg/mL), IL- 17F (pg/mL), IL-10 (pg/mL), TGF-b (pg/mL), IL-23 (pg/mL), IL-21 (pg/mL), IL-22 (pg/mL), IFN-g (pg/mL), IL-35 (pg/mL), IL-6 (pg/mL) were determined using commercially available enzyme linked immunosorbent assay (ELISA) kits according to the manufacturer’s instructions (Fine Test, Wuhan, China). The optical density was measured at 450 nm with an automatic ELISA reader (LT-4000 Microplate Reader, Labtech, Zambrow, Poland).

### 4.5. Statistical Analysis

The normality of distribution data were checked by the Shapiro–Wilk test. Correlations between variables meeting the criteria for parametric tests were analysed using Pearson’s correlation test. Correlations between miRNAs expression in serum and cytokines concentration/clinical outcomes/transcription factor expression in whole blood were conducted by the non-parametric Spearman test. Differences between analyzed groups were determined using the parametric *t*-test or non-parametric Mann–Whitney test, and for multiple comparisons Kruskal–Wallis with Dunn’s correction test. For all analyses, a value of *p* < 0.05 was considered significant. GraphPad Prism software ver. 8.4.2 was used to conduct analysis and prepare charts.

## 5. Conclusions

The study showed differences in the expression and correlation of the selected miRNAs between OA and RA and healthy subjects. Functional studies are necessary to prove the observed associations.

## Figures and Tables

**Figure 1 ijms-23-07925-f001:**
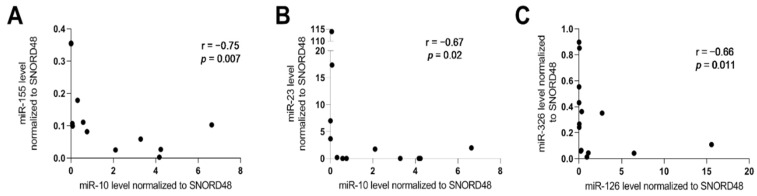
Correlation between analyzed miRNAs level in serum in RA patients. (**A**) Correlation between miR-10 and miR-155; (**B**) Correlation between miR-10 and miR-23; (**C**) Correlation between miR-126 and miR-326.

**Figure 2 ijms-23-07925-f002:**
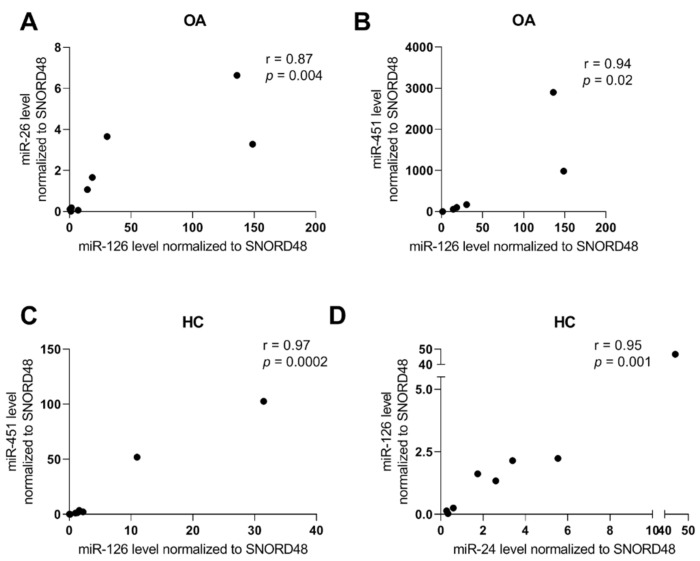
Correlation between analyzed miRNAs level in serum in OA patients (**A**,**B**) and healthy control subjects HC (**C**,**D**).

**Figure 3 ijms-23-07925-f003:**
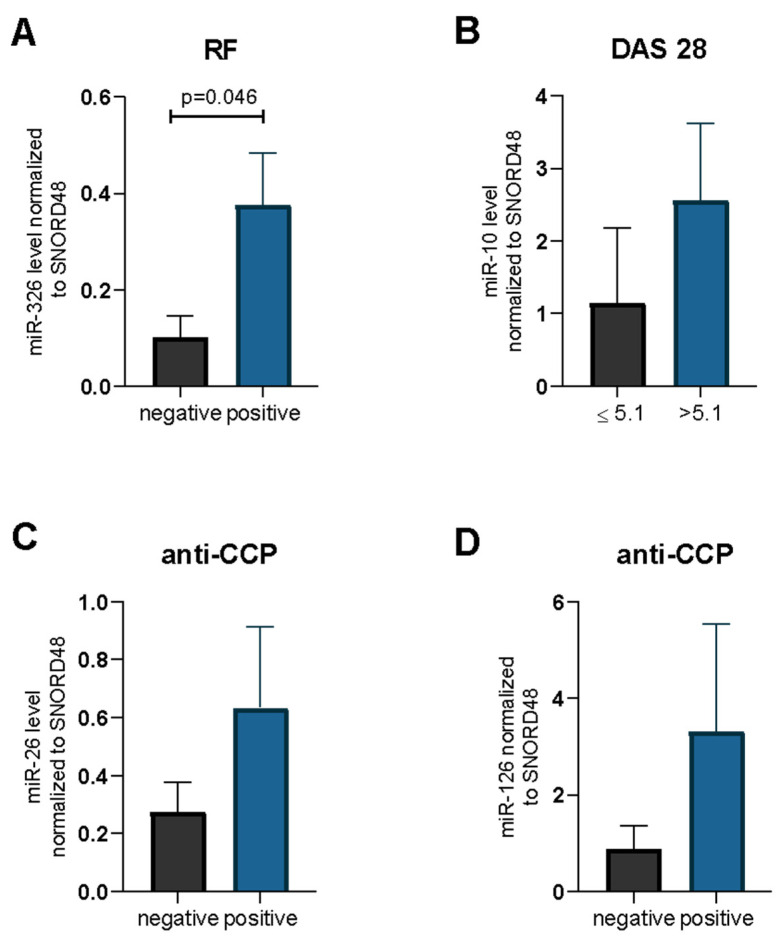
Rheumatoid factor (RF) (**A**), DAS-28 (**B**), anti-CCP (**C**) parameters in relation to miR-326 (**A**), miR-10 (**B**), miR-26 (**C**) and miR-126 (**D**) in patients with RA. Data are presented as mean ± SEM. miR-24, miR-26 miR-31, miR-146a, miR-155 expression levels.

**Figure 4 ijms-23-07925-f004:**
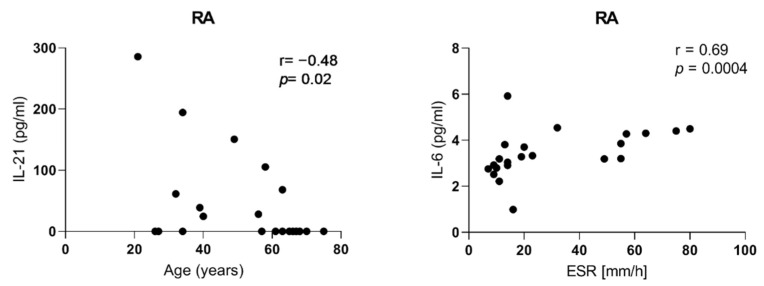
Correlation between IL-21 serum levels and age of RA patients, and between IL-6 serum levels and mean value of ESR in RA patients.

**Figure 5 ijms-23-07925-f005:**
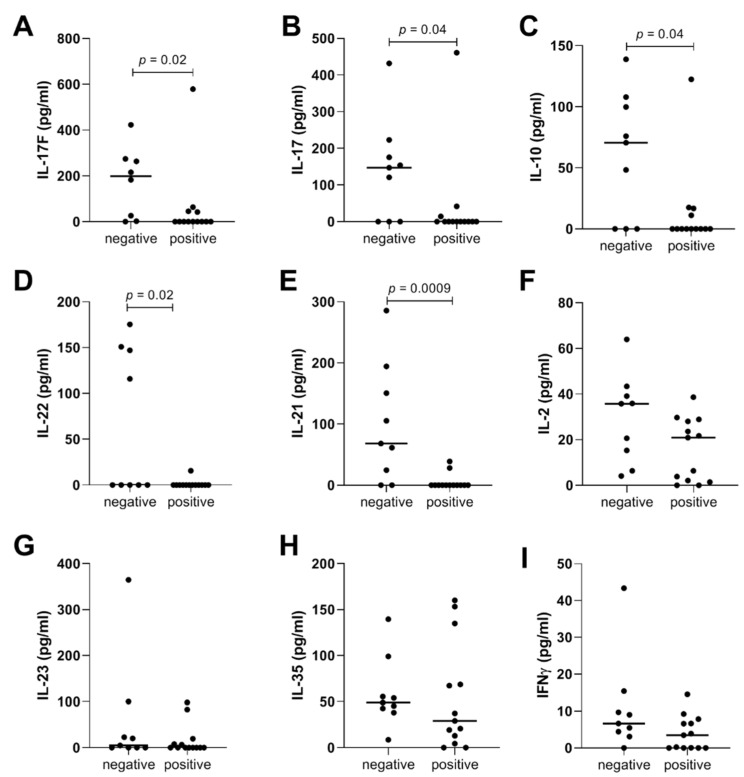
Serum cytokines concentration of (**A**) IL-17F; (**B**) IL-17; (**C**) IL-10; (**D**) IL-22; (**E**) IL-21; (**F**) IL-2; (**G**) IL-23; (**H**) IL-35; (**I**) IFNγ in RA patients with negative and positive Rheumatoid Factor (RF).

**Figure 6 ijms-23-07925-f006:**
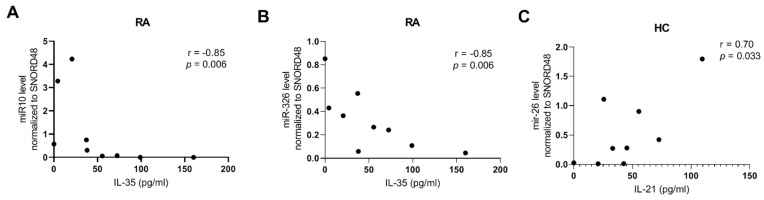
miRNA expression levels in serum and correlation between IL-35 in patients with RA (**A**,**B**) and IL-21 levels in HCs (**C**).

**Table 1 ijms-23-07925-t001:** Demographic and clinical characteristics of the all studies population.

Parameter	RA *n* = 23	OA *n* = 26	HC *n* = 29
age, years (median, range)	57.5 (21–75)	67 (28–85)	46 (41–63)
female/male	21/1	16/10	20/9
ESR—mm/h (mean ± SD)	29.86 ± 23.98	14.72 ± 8.369	-
CRP—mg/dL (mean ± SD)	21.68 ± 20.89	5.696 ± 1.396	-
Disease duration, years (median, range)	11.5 (0.5–21)	-	-
DAS-28 (mean ± SD)	4.993 ± 1.425	-	-
VAS (mm, mean ± SD)	58 ± 23.44	-	-
PLT (mean ± SD)	-	226.4 ± 52.88	-
vitamin D (mean ± SD)	-	31.83 ± 24.88	-
RF positivity, *n* (%)	13/9	-	-
anti-CCP positivity, *n* (%)	17/5	-	-
*Medication:*		-	-
cDMARDS: MTX/SSA/LEF, *n* (%)	7 (30%)	-	-
GKS + cDMARDS, *n* (%)	5 (22%)	-	-
antimalarials + cDMARDS, *n* (%)	5 (22%)	-	-
GKS + antimalarials, *n* (%)	1 (4%)	-	-
BIOLOGICS + cDMARDS, *n* (%)	3 (13%)	-	-
no drug, *n* (%)	1 (4%)	-	-
no data, *n* (%)	1 (4%)	-	-

“-”—no data; ESR—erythrocyte sedimentation ratio, CRP—C-reactive protein, DAS-28—disease activity score in 28 joints, VAS—visual analog scale, PLT—platelets, RF—rheumatoid factor, anti-CCP—anti-cyclic citrullinated peptide, cDMARDs—conventional disease-modifying antirheumatic drugs, MTX—methotrexate, SSA—sulfasalazine, LEF—leflunomide, GKS—corticosteroids.

**Table 2 ijms-23-07925-t002:** Demographic and clinical characteristics of the study population used for miRNA analysis.

Parameter	RA *n* = 12	OA *n* = 9	HC *n* = 9
age, years (median, range)	51 (21–75)	73 (28–80)	45.5 (44–60)
female/male	10/4	5/4	4/5
ESR—mm/h (mean ± SD)	35 ± 21.59	18.11 ± 9.02	-
CRP-mg/dL (mean ± SD)	24.92 ± 19.72	6.111 ± 1.764	-
Disease duration, years (median, range)	6.5 (0.5–18)	-	-
DAS-28 (mean ± SD)	5.018 ± 1.438	-	-
VAS (mm, mean ± SD)	64 ± 25.05	-	-
PLT (mean ± SD)	-	245 ± 52.97	-
vitamin D (mean ± SD)	-	37.07 ± 6.111	-
RF positivity, *n* (%)	13/9	-	-
anti-CCP positivity, *n* (%)	17/5	-	-
*Medication:*		-	-
cDMARDS: MTX/SSA/LEF, *n* (%)	2 (9%)	-	-
GKS + cDMARDS, *n*(%)	2 (9%)	-	-
antimalarials + cDMARDS, *n* (%)	3 (14%)	-	-
GKS + immunosuppressants, *n* (%)	1 (6%)	-	-
BIOLOGICS + cDMARDS, *n* (%)	2 (9%)	-	-
no drug, *n* (%)	1 (6%)	-	-
no data, *n* (%)	1 (6%)	-	-

“-”—no data; ESR—erythrocyte sedimentation ratio, CRP—C-reactive protein, DAS-28—disease activity score in 28 joints, VAS—visual analog scale, PLT—platelets, RF—rheumatoid factor, anti-CCP—anti-cyclic citrullinated peptide, cDMARDs—conventional disease-modifying antirheumatic drugs, MTX—methotrexate, SSA—sulfasalazine, LEF—leflunomide, GKS—corticosteroids.

## Data Availability

The data presented in this study are available on request from the corresponding author.

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
