# Peer review of "miR-10 and Its Negative Correlation with Serum IL-35 Concentration and Positive Correlation with STAT5a Expression in Patients with Rheumatoid Arthritis"

_ijms, 2022, doi:10.3390/ijms23147925_

Round 1

Reviewer 1 Report

The aims of presented studies is to correlated mico-RNAs with RA. Multi-factors depended rheumatoid arthritis (RA) and osteoarthritis (OA) are indeed required complex diagnosis tools. One of that was proposed by Authors od submitted manuscript – micro-RNAs level in sera of patients and control sera.

Before publication several point need to be setup.

Introduction part – line 30-32 – reference1 do not supported stated thesis i.e. RA and OA enhanced mortalities. Line 39 “ CD4+ T cells have a pivotal role in human’ health regulation” is not true and need to be conformed by references. Line 43- …”in the pathogenesis of RA, and other autoimmune diseases [3].”

Is that references corrected for statement? Above exemplars indicate that On Introduction part references and statements need to be carefully corrected.

In previous Authors publication (ref 15) total RNA was isolated from earlier sorted Th17 and Treg cells.

In current manuscript source of microRNAs were sera. Is that the same samples as already published? In Introduction proposal od sources of microRNAs in sera need to added.

Line 70 – 79 – critical for publication working hypothesis is missing and aims – ” microRNAs’ 75 expression profiles in serum on the RA” .  The explanation what Authors understood as expression profiles of micoRNAs.

Results part – are based on group of 23 RA patients, 26 OA patients and 29 healthy controls. Table 1 presented extensive characteristic of studied population. Studied group are very  heterogeneous as concern ages, sex, duration of disease, medical treatment. In order to build up more homogenous sample set  studied three group need to split for more then three. That is the major negative factor of presented studies –  not representative enough probe for conclusions presented. In Table 1 used abbreviation need to explain.  In Table 1 and 2 “- “ means no data or not detected?

MicroRNAs or miRNAs need to be used in whole manuscript as only in one form.

In conclusion panel of microRNA levels  – 10, 324, 26, 126 and 146 are correlated  with cytokines and signal transducers and  activator of transcriptions factors i.e.  (STAT3).  No rational explanation for presented ample of statistical correlation are provided.  It look like that Authors  try to correlated all

numbers and look for outcomes without any hypothesis of observed  up and down correlations.

Due to small number of  heterologous samples interesting observations are too early to be published.

Author Response

  1. Introduction part – line 30-32 – reference1 do not supported stated thesis i.e. RA and OA enhanced mortalities. Line 39 “ CD4+ T cells have a pivotal role in human’ health regulation” is not true and need to be conformed by references. Line 43- …”in the pathogenesis of RA, and other autoimmune diseases [3].”

Thank you very much for the Reviewer's kind comment. New, more precise references have been added to support that the RA and OA impact on enhanced mortality. Further sentences has been rewritten and supplemented with additional references

  1. Is that references corrected for statement? Above exemplars indicate that On Introduction part references and statements need to be carefully corrected.

Thanks’ to the Reviewer for the comment. It was corrected

  1. In previous Authors publication (ref 15) total RNA was isolated from earlier sorted Th17 and Treg cells.

This paper presents part of the our project results about selected miRNA antranscriptionon factors in RA and OA (aa s disease without autoimmune background). In the pro,ject we analyzed expression in serum, whole b,lood and additionally in sorted Th17 and Treg cells.

Due to the fact that our results and observations are large we published them in separated articles which we refer to: Th17/Treg-Related Transcriptional Factor Expression and Cytokine Profile in Patients With Rheumatoid Arthritis (Front. Immunol. 2020, doi:10.3389/fimmu.2020.572858) AND The interplay between transcriptional factors and micrornas as an important factor for th17/treg balance in ra patients. (Int. J. Mol. Sci. 2020, doi:10.3390/ijms21197169).

  1. In current manuscript source of microRNAs were sera. Is that the same samples as already published? In Introduction proposal od sources of microRNAs in sera need to added.

As the Reviewer suggested this information has been added

  1. Line 70 – 79 – critical for publication working hypothesis is missing and aims – ” microRNAs’ 75 expression profiles in serum on the RA” . The explanation what Authors understood as expression profiles of micoRNAs.

As the Reviewer suggested the explanation has been added.

  1. Results part – are based on group of 23 RA patients, 26 OA patients and 29 healthy controls. Table 1 presented extensive characteristic of studied population. Studied group are very heterogeneous as concern ages, sex, duration of disease, medical treatment. In order to build up more homogenous sample set  studied three group need to split for more then three. That is the major negative factor of presented studies –  not representative enough probe for conclusions presented. In Table 1 used abbreviation need to explain.  In Table 1 and 2 “- “ means no data or not detected?

We deeply appreciate the Reviewer’s professional remarks. The differences in clinical data are due to the fact that the patients came from different clinics and that is why the groups were heterogenous. We understand that some missing data may be annoying/confusing but we decided not to hide the facts. Perhaps some missing data may inspire readers to undertake similar analyses with our missing data in solving new emerging questions.

We have added abbreviations to the both tables. In Table 1 and 2 “- “ means no data

  1. MicroRNAs or miRNAs need to be used in whole manuscript as only in one form.

Thanks’ to the Reviewer for the comment. It was corrected

  1. In conclusion panel of microRNA levels – 10, 324, 26, 126 and 146 are correlated  with cytokines and signal transducers and  activator of transcriptions factors i.e.  (STAT3).  No rational explanation for presented ample of statistical correlation are provided.  It look like that Authors  try to correlated all numbers and look for outcomes without any hypothesis of observed  up and down correlations.

We deeply appreciate the Reviewer’s professional remarks. All analysis presented in this paper are part of our project associated with the SMADs/STATs signalling pathways. Our results were the subjects of  the described hypothesis based on the literature and observations. In previous papers we found significant association between expression level of mRNA and/or miRNAs/cytokines/clinical parameters in whole blood and sorted Th17/Treg cells. In the introduction chapter we briefly summarize these observations and refer to the our articles.

In present paper, we focused on serum and the expression profile of selected miRNA (associated with the SMADs/STATs signalling pathways) according to hypothesis that this molecules may be significant markers of disease and disease activity. Serum is valuable clinical material in this kind of analysis therefore it is easy to access and contrary to the analysis in PBMC no additional processing is needed to check the expression profile in serum.

  1. Due to small number of heterologous samples interesting observations are too early to be published.

Thank you very much for the Reviewer's kind comment. At this stage of the study, we could only report such results at the gene/miRNA expression level. Nevertheless, we are aware that further research is required and we are seeking funding to continue our research in the aspect of protein level but also functional mechanism. This information also has been added into the section where limitation of the study are presented (Discussion chapter).

Reviewer 2 Report

The manuscript entitled “miR-10 and its negative correlation with serum IL-35 concentration and positive correlation with STAT5a expression in patients with Rheumatoid Arthritis” by Agnieszka Paradowska-Gorycka and colleagues, investigates the potential association of selected miRNAs in serum with the expression of Th17/Treg transcription factors and clinical features in RA patients. Study was conducted on rheumatoid arthritis (RA), osteoarthritis (OA) and healthy subjects (HC).

This manuscript describes in context to the expression and analysis of miRNAs level in serum using LNA miRNA PCR assays and they found that mir-10 was detected only in RA patients. miR-10 expression was correlated with IL-35 serum concentration and mRNA level of STAT5a in whole blood in RA. Also, a tendency of the raised level of miR-10 was noted in RA patients with high activity disease. miR-326 was significantly upregulated in RA patients with rheumatoid factor presence. In case of HC the correlation between miR-26 and IL-21 serum levels and expression of SMAD3 have been found.

The authors have concluded that the differential association of transcription factor expression with serum miRNA levels may be important in the diagnosis and progression of RA and OA, but there are queries that need to be addressed.

General comments:

Ø  Authors in this study have performed mRNA expression of various genes in context to various signaling pathways, but I wonder how miRNAs were selected.

Ø  This study is based and is the continuation of their earlier research where they determine the correlation between transcription factor expression and Th17/Treg ratio. And this study was aimed to recognize the potential impact of the microRNAs’ expression profiles in serum on the RA onset and phenotype, and also on the selected transcriptional factors, SMAD3, STAT3, SOCS1 in RA, OA, and healthy controls (HCs) and they also studied the relationship between SMAD3, STAT3, SOCS1 and cytokine serum levels as well as examined cell-free circulating microRNAs, but I wonder authors have only measured the mRNA /miRNA levels in this context, I suggest authors should look at the protein expression/phosphorylation levels of the transcription factors as well.

Ø  Please describe properly how correlation between microRNAs level in serum in RA patients was performed?

Ø  Please re-write the introduction

Author Response

Ø  Authors in this study have performed mRNA expression of various genes in context to various signaling pathways, but I wonder how miRNAs were selected.

Thank you very much for the Reviewer's kind comment. Selected miRNA are associated with the SMADs/STATs signalling pathways. Present paper shows our results of miRNA expression profile in serum. However, these results are part of our larger project about SMADs/STATs signalling in Th17/Treg cells in RA and OA patients. We refer to the previous articles in the introduction chapter.  This information may have been unclear, in our previous version of the manuscript.

Ø  This study is based and is the continuation of their earlier research where they determine the correlation between transcription factor expression and Th17/Treg ratio. And this study was aimed to recognize the potential impact of the microRNAs’ expression profiles in serum on the RA onset and phenotype, and also on the selected transcriptional factors, SMAD3, STAT3, SOCS1 in RA, OA, and healthy controls (HCs) and they also studied the relationship between SMAD3, STAT3, SOCS1 and cytokine serum levels as well as examined cell-free circulating microRNAs, but I wonder authors have only measured the mRNA /miRNA levels in this context, I suggest authors should look at the protein expression/phosphorylation levels of the transcription factors as well.

We deeply appreciate the Reviewer’s professional remarks. At this stage of the study, we could only report such results at the gene/miRNA expression level. Nevertheless, we are aware that further research is required and we looking for funding to continue.

Ø  Please describe properly how correlation between microRNAs level in serum in RA patients was performed? – In the chapter of material and methods, section – statystical analysis, the method has been more precisely described.

Reviewer 3 Report

The authors present an interesting approach to chronic diseases through miRNAs

Indeed, these could represent a useful diagnostic and prognostic tool.

The correlation that seems to emerge with miRNAs linked to the inflammatory aspect is interesting; however, as pointed out by the authors, there are limits that should be better expressed:

- Even if the sample is limited, some aspects should be stratified or at least underlined:

. At least BMI, better body composition (it is known how obesity and overweight affect miRNA expression e.g. 10.2174 / 2211536608666181126093903)

. The level of physical activity should also be evaluated, as it is shown how this can affect the level of inflammation and therefore the quality of life (e.g. 10.3390 / jfmk6030058)

. Diet and the possible use of supplements also have an important impact on miRNAs, if it were possible you should have at least one report of your eating habits

Author Response

The authors present an interesting approach to chronic diseases through miRNAs

Indeed, these could represent a useful diagnostic and prognostic tool.

The correlation that seems to emerge with miRNAs linked to the inflammatory aspect is interesting; however, as pointed out by the authors, there are limits that should be better expressed:

- Even if the sample is limited, some aspects should be stratified or at least underlined:

. At least BMI, better body composition (it is known how obesity and overweight affect miRNA expression e.g. 10.2174 / 2211536608666181126093903)

. The level of physical activity should also be evaluated, as it is shown how this can affect the level of inflammation and therefore the quality of life (e.g. 10.3390 / jfmk6030058)

. Diet and the possible use of supplements also have an important impact on miRNAs, if it were possible you should have at least one report of your eating habits

We deeply appreciate the Reviewer’s professional remarks. Of course,

Round 2

Reviewer 1 Report

Manuscript  was signifcantly corrected. 

Reviewer 2 Report

The revised manuscript entitled miR-10 and its negative correlation with serum IL-35 concentration and positive correlation with STAT5a expression in patients with Rheumatoid Arthritis has been submitted with some changes incorporated.

It would be better if authors cite and add some 5-10 related references with recent advances on this subject. That will certainly add broader audience views.

Please check for plagiarism and typo's.

Grammatical errors should be corrected.

Reviewer 3 Report

The authors have made some changes but various problems remain that should be better underlined, perhaps by inserting a paragraph with the weaknesses of the manuscript:

- the impossibility of stratifying the population by stressing that this can be a confusing factor

- the choice of miRNAs should be better analyzed also in function of the possible influences of diet and lifestyle

- the conclusions should be broadened by suggesting in particular how a more accurate study could be made:

  a larger and more stratified sample

  choice of miRNAs according to possible "external" influences

  the action of the selected miRNAs and pathways involved

  more accurate choice of the control group

Round 3

Reviewer 3 Report

I appreciated all improvement done